# ENTAILMENT PROGRESSIONS: A ROBUST APPROACH TO EVALUATING REASONING WITHIN LARGER DISCOURSE

## ABSTRACT

Textual entailment, or the ability to deduce whether a proposed hypothesis is logically supported by a given premise, has historically been applied to the evaluation of language modelling efficiency in tasks like question answering and text summarization. However, we hypothesize that these zero-shot entailment evaluations can be extended to the task of evaluating discourse within larger textual narratives. In this paper, we propose a simple but effective method that sequentially evaluates changes in textual entailment between sentences within a larger text, in an approach we denote as *"Entailment Progressions"*. These entailment progressions aim to capture the inference relations between sentences as an underlying component capable of distinguishing texts generated from various models and procedures. Our results suggest that entailment progressions can be used to effectively distinguish between machine-generated and human-authored texts across multiple established benchmark corpora and our own EP4MGT dataset. Additionally, our method displays robustness in performance when evaluated on paraphrased texts a technique that has historically affected the performance of well-established metrics when distinguishing between machine generated and human authored texts.

## 1 INTRODUCTION

As Large Language Models (LLMs) expand and evolve to accommodate more complex language generation tasks (e.g., significant advances in machine translation (Lai et al., 2023), logical reasoning (Liu et al., 2023), summarization (Zhang et al., 2023), complex question answering (Tan et al., 2023)), we are witnessing a growing number of machine-generated text (MGT) in both online and offline environments.[1] This, in turn, has raised concerns regarding authenticity and regulations,[2,3] drawing attention to MGT detection as both a safeguard and qualifier for authentic human authorship, which has become quite a hot topic in Natural Language Processing (NLP).[4]

Intuitively, machine-generated texts can display lexical, syntactic, and semantic properties that are distinguishable from human authored texts, potentially guiding MGT detection implicitly, as a latent property, or explicitly as a directly encoded feature (Georgiou, 2024). For example, MGT detection methods like entropy and log-likelihood, which assess the probability of a text being machine generated based upon individual token probabilities encoded by a given LLM, take into account how LLMs functionally operate as next word predictors (He et al., 2023). Thus, evaluating where LLMs situationally differ from human authorship in relation to both their observed behaviour and functionality can expand the scope of feature selection within MGT detection to capture these differences more effectively and in a more interpretable manner.

Textual entailment, or the relationship between a given premise and its potentially inferred hypothesis, has been previously used to evaluate how LLM text generation differs from human authorship in regard to an LLM's ability to generate text in accordance with prior informational constraints

---

[1]For a comprehensive overview of LLM capabilities see Guo et al. (2023) and Chang et al. (2024).

[2]https://eur-lex.europa.eu/legal-content/EN/TXT/?uri=celex%3A52021PC0206

[3]https://www.whitehouse.gov/ostp/ai-bill-of-rights/

[4]For an in-depth analysis of the task, existing corpora and detection methods, see Wu et al. (2023).

(Dagan et al., 2022). In areas like question answering and dialogue systems, calculating the textual entailment between a prior conversation and a machine-generated response can examine whether a model produces relevant and accurate text, a behaviour assumed to be exhibited in human authorship and communication (Ben Abacha & Demner-Fushman, 2019; Dziri et al., 2019). Based on observations of differences in textual entailment between MGTs and human-authored texts in relation to prior conversations, an interesting question arises: *can textual entailment be directly encoded and utilized as a feature for MGT detection?*

In this paper, we:

**(1) Introduce *entailment progressions***, a framework in which a given piece of text can be represented as a series of values, with each value representing the level of textual entailment between sentences in a text. These entailment progressions aim to measure the extent to which a model generates each individual utterance in logical reference to its previously generated utterances (i.e., identifying how new information is introduced in relation to the preceding content: *in support*, *in contradiction*, or with *no relation* (neutral)). We believe that entailment progressions provide a unique perspective and should be considered in qualifying LLM behaviour to achieve a more in-depth analysis.

**(2) We propose a novel benchmark dataset, `EP4MGT` (Entailment Progressions for Machine Generated Text)**, comprising 70,158 machine-generated responses across eight state-of-the-art LLMs.[5]

## 2 RELATED WORK

The definition of recognizing textual entailment (RTE) as outlined by Dagan et al. (2005) and later expanded upon by Korman et al. (2018) is as follows: *"a text T textually entails a hypothesis H relative to a group of end users G just in case, typically, a member of G reading T would be justified in inferring the proposition expressed by H from the proposition expressed by T"*. This definition incorporates three key aspects of RTE. First, it does not require any knowledge beyond the justifiable inference that can be made between a given text and its hypothesis Feldman (2003). Second, this justifiable inference is subject to the characteristics exhibited by a group of end users G, in which users outside of this group may differ in their inferences due to personal factors that may influence how they interpret logical relationships Bos & Markert (2005). Third, the logical component of entailment is textually constrained, rendering it dependent on linguistic factors such as grammar, semantic, and syntactical choices Braun (2001).

Current RTE modelling approaches require two main steps. First, the features of premise T and hypothesis H are extracted in order to represent the statements in accordance with relevant linguistic mechanisms associated with textual entailment. Second, the statements are fed into a supervised multi-class classification model which predicts whether a premise-hypothesis pair possesses *positive* (the hypothesis can be inferred to be true if the premise is true), *negative* (the hypothesis can be inferred to be false if the premise is true), or *neutral* (the hypothesis' truth is not sufficiently conditional upon the premise being true) entailment. For an in-depth overview of RTE resources, approaches, and applications, see Putra et al. (2024).

## 3 METHODOLOGY

### 3.1 HYPOTHESIS

We incorporate Korman's RTE approach into the task of detecting MGT under the premise that determining inference relations between sentences in a text is a component of identifying authentic human authorship.

Take, for example, a short story written by ChatGPT. While the story may contain relevant content pertaining to the subject matter and utilize vocabulary similar to its human counterpart, ChatGPT may employ a more simplistic narrative structure without the stylistic nuance or variability typical of human authors. While these LLMs are autoregressive models that generate the next token based

---

[5]The dataset will be made freely available to the research community upon acceptance.

on the previous sequence (without explicitly modelling entailment in the process), our interest lies in exploring whether certain logical patterns are internally captured to some degree.

Regardless of the manner in which these evaluations are conducted, the structure of a textual narrative (like a short story) is an identifiable linguistic feature that can be used to distinguish between texts. We posit that in settings where texts must be logically structured to advance a given claim or narrative purpose, sentence-level evaluation can identify and distinguish structural differences between different generative processes. This process involves examining the inference relations between a new sentence and its overarching premise, as well as between sentences within the text. RTE models can determine the probabilities of entailment, contradiction, and neutrality between a sentence and its preceding text (to identify how the sentence logically corresponds to prior context). These probabilities can be then assembled into *"entailment progressions"*, which are vectors composed of sequentially calculated probabilities of inference relations between a given sentence and the sentences preceding it.

The formal definition of the entailment progressions of a given text can be expressed as follows:

$$EP_{3 \times n} = \begin{bmatrix} c_0 & c_1 & \cdots & c_{n-1} \\ p_0 & p_1 & \cdots & p_{n-1} \\ n_0 & n_1 & \cdots & n_{n-1} \end{bmatrix}$$

where *EP* is a matrix composed of *c*, *p*, *n* row vectors representing the contradiction, positive, and neutral entailment probabilities between a sentence at a chosen index and its prior sentences in a given text. To compute these values at a given point in a text, we introduce the following equations:

$$EP_{0,i} = C(s_{i+1-w:i+1}, s_{i+1})$$
$$EP_{1,i} = P(s_{i+1-w:i+1}, s_{i+1})$$
$$EP_{2,i} = N(s_{i+1-w:i+1}, s_{i+1})$$

where *E* represents the model used for calculating entailment between a sentence *s* at a given point in the text *i* and the sentences preceding it within a context window of size *w*.

Motivated by observed discourse phenomena, such as the referential connection between (summarizing) titles and the sentences in their corresponding texts, as well as between sentences in close proximity (Mirkin et al., 2010), entailment progressions use entailment as a heuristic for identifying logical relationships between key components of a text. Given this emphasis on the logical relation between a chosen sentence and its overarching premise (i.e., a title), we also include the following equations:

$$EP_{0,i} = C(p, s_i)$$
$$EP_{1,i} = P(p, s_i)$$
$$EP_{2,i} = N(p, s_i)$$

where *E* represents the model used for calculating entailment between the general premise defining the full text *p* and a sentence or collection of sentences *s*.

Based on our analysis of existing RTE literature, we hypothesize that if the logical relationships between components of a text are distinguishable linguistic features that underlie a set of texts produced by either models or humans, and if entailment progressions effectively represent this set of relationships, then entailment progressions can be used to identify the source of a set of texts. Our hypothesis hinges upon two interconnected inquiries: *Are entailment progressions a meaningful feature of a text?* And, if so, *is the governing structure of these logical relationships reproducible across texts produced by the same author?* We suggest that our hypothesis can be validated by evaluating whether entailment progressions can serve as a feature for identifying and interpreting human authorship. If we can identify MGTs using only their entailment progressions, this would experimentally confirm that they are both meaningful and reproducible features across texts generated through the same procedure.

## 3.2 DATASETS

We conduct our experiments on two freely available English corpora from previous studies and one newly created dataset.

**MULTITuDE.** This dataset includes 74,081 texts (comprising 7,992 human-written and 66,089 machine-generated texts), distributed across 11 languages (Macko et al., 2023).[6] The human-written portion of the corpus consists of news articles from the MassiveSumm dataset (Varab & Schluter, 2021). The authors used the titles of the human-written articles for prompting eight different LLMs to generate the corresponding MGTs.

**Ghostbuster.** This corpus includes both human-authored and `ChatGPT`-generated text across three domains: creative writing, news, and student essays (Verma et al., 2023). The creative writing collection is sourced from the `/r/WritingPrompts` subreddit and contains both the original prompts and the corresponding MGT/human-authored texts. The human written collection for the news dataset is based on the Reuters 50-50 authorship identification dataset (Houvardas & Stamatatos, 2006), while the student essay dataset contains high school and university-level essays collected from IvyPanda.[7]

In order to bypass the fixed structure of some of these texts (e.g., news articles), while also covering a diverse set of topics, we build a new dataset, `EP4MGT`, through which we aim to assess the differences in structure between human-authored and MGTs, specifically within the context of online debates and discussions.

**EP4MGT.** We draw the human-authored texts from the `CMV` dataset (Tan et al., 2016), which consists of user interactions from the `/r/ChangeMyView` subreddit. This Reddit community features posts in which a user presents their original beliefs and rationales, challenging others to contest these viewpoints.[8] Given a title from the `CMV` dataset, we task the following LLMs: `ChatGPT`, `GPT4` (Achiam et al., 2023), `Gemini` (Team et al., 2023), and `Mistral` (Jiang et al., 2023) (`mixtral-8x7b`, `mistral-7b`, `mistral-small`, `mistral-medium`, `mistral-large`) with writing an argument (either in favour or against the topic) consisting of at least seven sentences. The prompt used for generating the `EP4MGT` dataset is presented in Figure 3. Detailed dataset statistics are presented in Table 2.

It is important to note the varying sentence lengths (and by extension varying word counts) of the texts included in these corpora. In order to prevent sentence length being a confounding factor in our analysis, we removed both human-authored and machine-generated texts that were outliers in their respective sentence length distributions (e.g., texts containing only one or two sentences, groups of texts that contained fewer than 50 instances of a specific length).[9]

## 3.3 Experimental Design

To ensure that our hypothesis is satisfied, we design an experimental setup that effectively accounts for potential confounding limitations that may arise during analysis.

First, in order to establish a fair comparison between a set of human-authored and machine-generated texts, both sets must *"further the same logical premise"* and pertain to the same language generation task. This effectively controls for style (e.g., news articles, social media discourse, persuasive essays) that could otherwise overemphasize the differences in entailment progressions between human-authored and model-generated texts.

Second, the texts under examination must be preprocessed in a way that removes any textually confounding identifiers that can further accentuate comparative differences in entailment progressions. This process involves removing any elements within the text that are not relevant to the narrative at hand. These elements include, but are not limited to, the language in which the texts are written, identifiable markers from the media sources (e.g., platforms like Reddit include identifiable tags), and anomalies in sentence length. This helps ensure that the analysis focuses solely on the content of the text.

---

[6]For the purpose of our analysis, we selected only the English subset of the dataset.

[7]As the authors did not have access to the original news headlines or essay prompts, they used ChatGPT to generate headlines and prompts before creating the corresponding articles and essays.

[8]The dataset can be found at: `https://convokit.cornell.edu/documentation/winning.html`.

[9]Figure 4 shows the distribution of the number of sentences across the various models in the corpora used in this study, while Table 2 presents an overview of the filtered and unfiltered corpora.

When controlling for these conditions, we design an experimental setting that is suitable for determining whether entailment progressions can be effectively used as a feature for assessing human and model authorship. This setting involves calculating the entailment progressions for texts from both human-authored and model-generated sets, and then training a classification algorithm to distinguish between the two sources. If the algorithm performs well on the classification task, then we can assume that entailment progressions are a viable feature for differentiating between machine-generated and human-authored texts.

Based on our hypothesis (cf. Section 3.1), we propose two key approaches for constructing the entailment progressions. The first approach (denoted *"Title-Sentence"*) involves calculating the entailment between the general premise of the text and the sentences within the text. This approach assesses the logical relationship between each sentence and the premise it (is attempting to) support. The second approach (denoted *"Sentence-Sentence"*) involves calculating the entailment between a given sentence and its preceding context. This method uses a sliding context window, examining a given number of sentences (based on the selected window size) directly prior to the evaluated sentence.

In line with the experimental design previously outlined, we generated the *Sentence-Sentence* entailment progressions using context window sizes of 1, 2, and 3 sentences for all datasets. Regarding *Title-Sentence* entailment progressions, as we do not have the general premise for the `MULTITuDE` and `Ghostbuster` datasets, we only generate it for the `EP4MGT` dataset. In this case, the general premise is the title of the original human-authored CMV post, which we used to generate the LLM responses addressing the argument conveyed by the title.

While most of the existing datasets (e.g., SNLI (Bowman et al., 2015), MNLI (Williams et al., 2017)) address the RTE task at sentence-level, logical connections can go beyond consecutive sentences. As such, we rely on DeBERTa pretrained on eight RTE datasets, including DocNLI (Yin et al., 2021), a dataset spanning various lengths for both premises and hypotheses. For performing the experiments, we relied on the HuggingFace transformers library (Wolf et al., 2020).[10] To test our hypothesis, we trained multi-layer perceptrons (MLPs) with a single hidden layer on these entailment progressions to classify texts within a dataset as either model-generated or human-authored. It is important to note that when assembling the training and testing datasets for the MLP models, we only selected entailment progressions that met the same conditions (e.g., *Sentence-Sentence* entailment progressions with a context window size of 2 sentences).[11] Since the entailment progressions vary in length and are sequential, we leveraged a Time Series MLP implementation available through tslearn,[12] a Python package dedicated to time series modelling and machine learning.

## 4 RESULTS AND DISCUSSION

In Figure 1 we showcase two MGTs from the `EP4MGT` dataset. Although these two MGTs are generated by different models (i.e., `GPT4` and `mistral-large`), pertain to different subject matters, and display low textual similarity (0.0718 as calculated using SentenceBERT (Reimers, 2019), a modified BERT that derives semantically sentence embeddings that can be compared using cosine similarity), they exhibit high entailment progression similarity (5.9948 using Dynamic Time Warping distance, that measures the similarity between time series (Müller, 2007)) between each other.

Table 1 highlights the performance of our MLP model when trained solely on various types of entailment progressions across the `EP4MGT`, `MULTITuDE`, and `Ghostbuster` corpora. In our analysis of the two approaches for constructing entailment progressions, we observe that the *Title-Sentence* approach generally underperforms in the `EP4MGT` dataset. For the `EP4MGT` dataset, in terms of $F_1$ score, the performance drop ranges from 13% to 21% when comparing the *Title-Sentence* approach to the *Sentence-Sentence* approach with a one-sentence context window (CONTEXT-1), to two (CONTEXT-2) and three-sentence (CONTEXT-3) context windows, respectively. While the three-sentence context window approach consistently outperforms other entailment progression methods

---

[10]`https://huggingface.co/MoritzLaurer/DeBERTa-v3-base-mnli-fever-docnli-ling-2c`
[11]We perform a binary classification task between human-authored texts and texts generated by a specific LLM (e.g., GPT4).
[12]`https://tinyurl.com/TimeSeriesMLPClassifier`

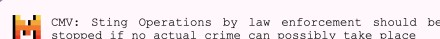

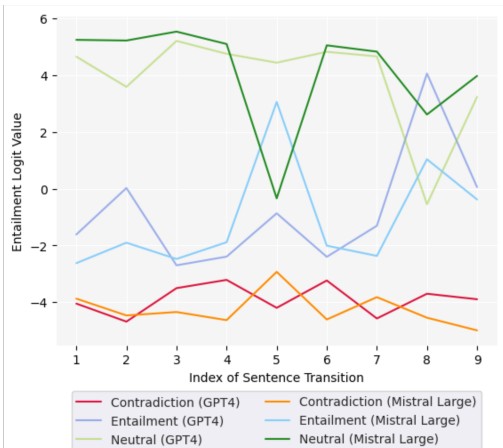

Figure 1: Examples from the EP4MGT dataset displaying low semantic similarity and high entailment progression similarity.

in the EP4MGT dataset, this trend does not hold for the MULTITuDE and Ghostbuster datasets, where the best performing method depends on both the model and the narrative scenario. Overall, the results show that entailment progressions capture aspects of the evaluated text that can help models (like MLP) to identify human authorship, highlighting the potential insights entailment progressions could provide through further exploration.

Table 1: Macro $F_1$ scores for *Title-Sentence* and *Sentence-Sentence* (using context window sizes of 1, 2, and 3 sentences) entailment progressions across the EP4MGT, MULTITuDE, and Ghostbuster corpora.

| EP4MGT | ENTAILMENT + MLP | | | | MULTITuDE | ENTAILMENT + MLP | | | GHOSTBUSTER | ENTAILMENT + MLP | | |
|---|---|---|---|---|---|---|---|---|---|---|---|---|
| | TITLE-SENTENCE | CONTEXT-1 | CONTEXT-2 | CONTEXT-3 | Δ | | CONTEXT-1 | CONTEXT-2 | CONTEXT-3 | | CONTEXT-1 | CONTEXT-2 | CONTEXT-3 |
| GPT4 | 0.681 | 0.832 | 0.896 | **0.903** | -0.046 | vicuna-13b | 0.786 | **0.839** | 0.827 | Claude | **0.827** | 0.804 | 0.776 |
| ChatGPT | 0.743 | 0.892 | 0.979 | **0.979** | -0.008 | llama-65b | 0.570 | 0.659 | **0.663** | GPT3.5-turbo | 0.922 | **0.922** | 0.911 |
| gemini-1.0-pro | 0.681 | 0.818 | 0.897 | **0.902** | -0.031 | GPT4 | 0.784 | **0.857** | 0.841 | GPT3.5-turbo − prompt 1 | 0.834 | 0.825 | **0.837** |
| mistral-7b | 0.735 | 0.825 | 0.911 | **0.915** | -0.001 | GPT3.5-turbo | 0.768 | 0.810 | **0.811** | GPT3.5-turbo − prompt 2 | 0.909 | **0.917** | 0.871 |
| mistral-small | 0.695 | 0.834 | 0.939 | **0.940** | -0.042 | text-davinci-003 | 0.720 | 0.704 | **0.750** | GPT3.5-turbo − writing | **0.926** | 0.920 | 0.921 |
| mistral-medium | 0.718 | 0.869 | 0.935 | **0.939** | -0.054 | alpaca-lora-30b | **0.696** | 0.657 | 0.669 | GPT3.5-turbo − semantic | **0.956** | 0.906 | 0.902 |
| mistral-large | 0.710 | 0.869 | 0.932 | **0.945** | -0.015 | opt-66b | 0.524 | 0.661 | **0.690** | | | | |
| mixtral-8x7b | 0.723 | 0.845 | 0.935 | **0.936** | -0.011 | opt-iml-max-30b | 0.588 | **0.768** | 0.767 | | | | |

Similar to recent work leveraging paraphrasing as a means of evaluating the robustness of different MGT detection approaches (Verma et al., 2023), we also examine the change in performance exhibited by our MLP model when trained on the entailment progressions of the paraphrased texts (where Δ = best model $F_1$ - best model paraphrased $F_1$). For this, we leveraged the same methodology as Verma et al. (2023) and Chakraborty et al. (2023), in which each sentence is individually paraphrased using the Pegasus transformer model (Zhang et al., 2020). When trained on the entailment progressions of the paraphrased texts from the EP4MGT dataset, the model exhibits a performance degradation of up to 5% in terms of $F_1$ score. In addition to these scores, Figure 2 illustrates the changes in between the mean positive entailment progressions for the EP4MGT dataset and its paraphrased counterpart.

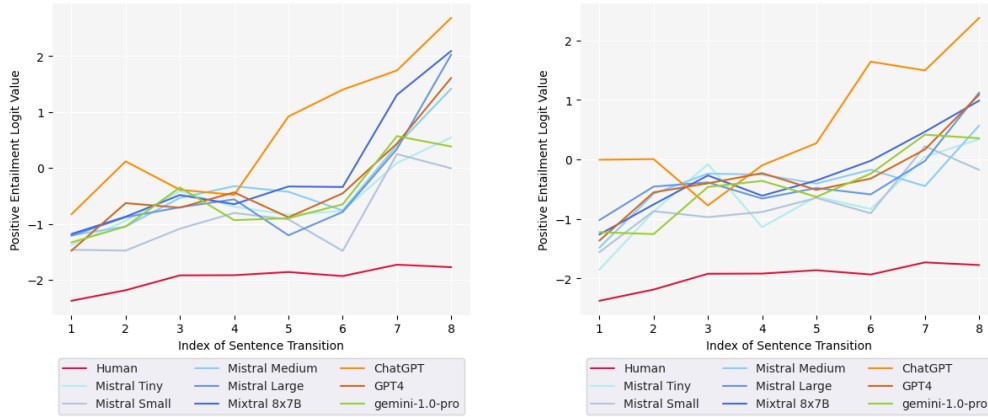

Figure 2: Mean positive entailment progressions of texts from `EP4MGT` dataset before (left) and after (right) paraphrasing.

## 5 CONCLUSION

In this paper, we introduce entailment progressions, a novel representation of the underlying logical structures of textual narratives for identifying human and model authorship. We also present `EP4MGT`, a dataset specifically designed to evaluate the logical approaches of humans and those produced by a suite of state-of-the-art LLMs, highlighting new avenues for exploring the properties and scope of entailment progressions as a latent descriptor of authorship.

Given that entailment progressions can be generated from any multi-sentence text, their potential applications could extend to the broader area of text attribution, thus providing insights in their utility as identifiers of authorship (be it human or model-based). This would also position our framework alongside more traditional lexical, syntactic, and semantic descriptors of style.

In future work, we plan on examining the effectiveness of entailment progressions in other experimental settings, across different languages, tasks, and genres. Although through our framework we have successfully detected MGTs in several English corpora with fixed narrative structure (i.e., personal claims, news articles), testing entailment progressions on datasets in languages with different underlying logical conventions or genres without any explicit logical constraints could reveal broader applicability.

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

## A  APPENDIX

The prompt used for generating the `EP4MGT` dataset:

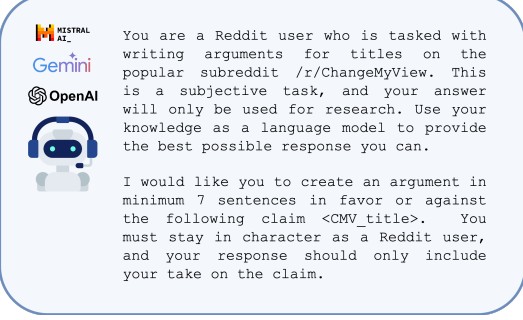

Figure 3: The prompt used for generating the `EP4MGT` dataset.

The distribution of the sentence counts across the various models in the corpora used in this study is presented in Figure 4, while Table 2 shows the number of samples in the corpora.

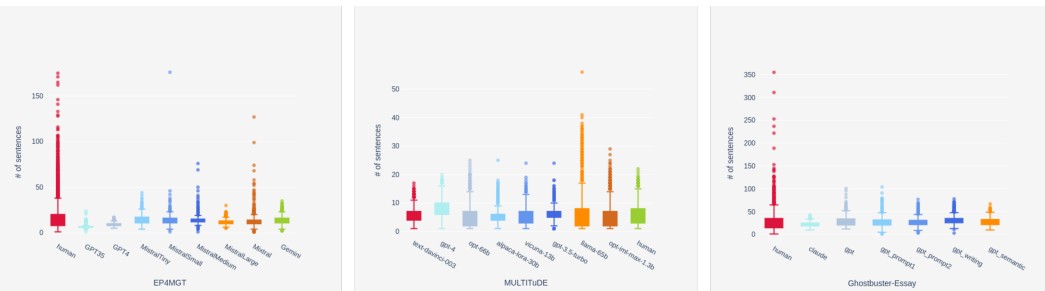

Figure 4: Distribution of number of sentences across the various models in the corpora used in this study.

Table 2: Number of machine-generated and human-written texts in the corpora.

| DATASET | MODEL | TOTAL # OF SAMPLES | # OF SAMPLES USED |
|---|---|---|---|
| EP4MGT | GPT4 | 3,658 | 3,658 |
| | ChatGPT | 10,000 | 9,928 |
| | gemini-1.0-pro | 6,500 | 5,868 |
| | mistral-7b | 10,000 | 10,000 |
| | mistral-small | 10,000 | 8,663 |
| | mistral-medium | 10,000 | 10,000 |
| | mistral-large | 10,000 | 10,000 |
| | mixtral-8x7b | 10,000 | 10,000 |
| | human-written | 10,000 | 3,864 |
| MULTITuDE | vicuna-13b | 3,298 | 982 |
| | llama-65b | 3,288 | 764 |
| | GPT4 | 3,300 | 1,828 |
| | GPT3.5-turbo | 3,300 | 1,262 |
| | text-davinci-003 | 3,300 | 1,056 |
| | alpaca-lora-30b | 3,297 | 749 |
| | opt-66b | 3,293 | 755 |
| | opt-iml-max-30b | 3,287 | 707 |
| | human-written | 3,097 | 1,006 |
| Ghostbuster | claude | 1,000 | 958 |
| | GPT | 1,000 | 920 |
| | GPT-prompt 1 | 1,000 | 884 |
| | GPT-prompt 2 | 1,000 | 899 |
| | GPT-writing | 1,000 | 910 |
| | GPT-semantic | 1,000 | 955 |
| | human-written | 1,000 | 730 |

