# OpenReview forum: "Entailment Progressions: A Robust Approach to Evaluating Reasoning Within Larger Discourse"
_ICLR.cc/2025/Conference — Submitted to ICLR 2025_

### Official Review · Reviewer_GjwN · 2024-10-20

**Soundness:** 2
**Presentation:** 3
**Contribution:** 2
**Rating:** 3
**Confidence:** 5

**Summary:**

The paper describes a new idea to model texts in terms of entailment progressions for the purpose of the deciding that a text is generated by a human or by AI. Entailment progression is measured by aggregating entailment relations using DeBERTa between titles as a premise and following sentences or between a sentence and preceding sentences. The approach is tested against 3 datasets crested for the purpose of detecting the source as human or AI. The third dataset is created by the authors using the subreddit ChangeMyView by taking the post and having LLMs generate a defending or opposing text of 7 sentences. The authors observe high succes for a simple MLP trained on the data when using 3 sentence contexts in the sentence-sentence method of modeling. Performance is similar across models.

**Strengths:**

- a new concept of text entailment progression is described and implemented using DeBERTa’s entailments classification
- the use of this modeling of entailment for human-ai source classification
- dataset EP4MGT with human and AI generated texts defending or opposing a post on ChangeMyView subreddit, with AI texts generated by a number of SOTA LLMs.
- testing and analysis of the performance of a MLP trained and tested.

**Weaknesses:**

- the central idea that AI uses different entailment relations than humans is not motivated nor deeply analysed in the results and discussion. Why would AI generate a different pattern and what type of differences are these?
- The entailment progression is not compared against other approaches e.g. simply train a model for binary classification on implicit or explicit features. What does the entailment progression bring that other approaches do not?
- entailment progression as such is an interesting idea but it should be tested directly first to see to what extend it captures distinctive patterns and what these patterns represent as such. News articles e.g. follow certain patterns starting the main story, followed by different viewpoints, more details and conclusions. How is this reflected in the entailment progression?
- AI is know to use particular phrasing (e.g. being positive and constructive) regardless of the topic. Is this reflected in the entailment progression?
- Why did not you do a cross-database test to see if the entailment progression representation abstracts from the data as such and generalises?
- The discussion of the results focuses a lot on the scores and does not give in any insights into what different patterns are detected. The authors should include a section describing the entailment progression patterns as such across the data sets.

**Questions:**

- clarify better why this idea should work in the first place
- test the entailment progression representation as such independently from the task

---

### Official Review · Reviewer_tGAd · 2024-10-30

**Soundness:** 2
**Presentation:** 3
**Contribution:** 2
**Rating:** 5
**Confidence:** 3

**Summary:**

The paper proposes that human-written and LLM-written texts can be distinguished by
computing (textual) entailment between every sentence and the previous one in text,
called "entailment progressions" and then feed this information to a classifier. It works to an extent.

**Strengths:**

The paper presents a simple idea and carries out a reasonable experiment on multiple corpora to test its validity.

**Weaknesses:**

* The idea that entailments should be computed between each sentence and its directly preceding sentence seems, at best, an oversimplification that assumes that texts are structured sequentially. However, there is a decade-long body of work from at least
two areas, namely discourse analysis, and argumentation analysis, which shows that texts (just like sentences) have substantially
more structure than just sequential one; typically discourse structure is modelled hierarchically, with tree structures: compare
the Penn Discourse Treebank; compare Rhetorical Structure Theory; compare the Toulmin model of argumentation, for example. None of these points of comparison is mentioned in your paper, but should be.

* Consequently, in many cases it does not really make sense from a theoretical point of view to even consider whether entailment holds between two adjacent sentences. Of course, this does not preclude that 'entailment progression' in practice is a useful heuristic -- but its (lack of) theoretical status should be clearly pointed out, and presented not as currently in a vaccum, but before the backdrop of relevant work on discourse structure.

* The experimental setup in the paper is rather artificial, set up in a manner to showcase the usefulness of the 'entailment progression'.
   Compare l. 203-209: (al)most properties of the texts are thrown away. Only results for using entailment progression as a predictor are shown in the results table (Table 1); no baselines are presented or shown, so it is extremely hard to put these numbers into perspectives. This setup does not convince me that entailment progression is the most useful representation. I would be considerably more convinced if the paper showed how badly alternative approaches based on style, on sentence length, etc. (as listed in that paragraph) would do.

* Table 1 is so small as to be all but unreadable in a printed version. Please increase the font size and/or split the table appropriately.

**Questions:**

See Weaknesses above.

---

### Official Review · Reviewer_hAbn · 2024-11-03

**Soundness:** 2
**Presentation:** 2
**Contribution:** 2
**Rating:** 3
**Confidence:** 4

**Summary:**

This paper presents a new idea for predicting the authorship of a text (human-authored vs. machine-authored), using a novel approach called "entailment progressions". The approach is to calculate a characteristic matrix corresponding to a text, in which entailment probabilities are stored between each sentence and all preceding sentences in the text. This matrix can be used as an input feature vector to an authorship classifier. The paper demonstrates a simple MLP which is trained on a corpus of such data and tested on heldout data, repeated in several settings across 3 datasets. The authors promise to release their new dataset upon acceptance.

**Strengths:**

1. The paper presents a novel idea for explicitly characterizing the structure of text (human- or machine-generated) in terms of entailment relations between each sentence in the discourse and each earlier sentence. This idea is reminescent of argument mining techiques, like argument relation identification [1].

2. The paper presents a new dataset, EP4MGT, which is designed to test models' ability to distinguish human- from machine-generated texts. Furthering our understanding of this would be very valuable for the field.


[1] Debela Gemechu, Ramon Ruiz-Dolz, and Chris Reed. 2024. ARIES: A General Benchmark for Argument Relation Identification. In Proceedings of the 11th Workshop on Argument Mining (ArgMining 2024), pages 1–14, Bangkok, Thailand. Association for Computational Linguistics.

**Weaknesses:**

1. The idea of entailment progressions could be better motivated. I am left wondering, "why should such a method work well at all" in distinguishing machine-generated text, especially when samples are small and could plausibly have high variability in argument structure? Even just a few sentences explaining this would be helpful. I am a bit concerned that scores vary so much by dataset, model, and test setting, in table 1. This makes me doubt the conclusions of the paper, and there is little discussion of the results to help clarify.

2. The results in Table 1 on EP4MGT look initially promising but are a bit difficult to interpret (and seem to disagree somewhat with results on the other datasets).
	- Shown is macro F1 scores for an MLP trained on different datasets (each generated by a certain model). It appears that the scores are not necessarily comparable down each column, but only across each row within a dataset. I assume this because the macro F1 score averages the precision/recall on a model's generations mixed in with the single set of human generations (and the human generations must be reused across all models?). This would mean that not only are the datasets different across models, but the class balances are also different. This is probably being obscured by the averaging in macro F1.
	- It would be nice to have some discussion explaining the results of table 1 in more detail, since there are many models and three different datasets to analyze. I cannot find discussion of results across the three datasets, which would be helpful since a very different set of models was used on each dataset. Also, why is this? It would be good to compare a few models consistently across the datasets.

3. Both Figure 1 and 2 show graphs with axes that I cannot find an explanation for. In general, the score calculation across the paper was not very clear. I would also point out that the equations which describe the feature vectors in L130-132 seem to set the same variables as in L142-144 (perhaps I am misreading this? the mathematical notation used is a little confusing)

4. The writing is mostly clear, but sometimes too high-level and feels like a rough sketch. Much more motivation and analysis is needed for the experimental results, and I am hopeful that a revision of this work will yield an interesting paper.

**Questions:**

1. Can you give more detail about how scores in table 1 are calculated? I notice some simple things, such as ChatGPT scoring higher than GPT-4 on EP4MGT, which seems like a red flag.
2. Why are models performing so inconsistently between datasets?

---

### Official Review · Reviewer_SsW6 · 2024-11-04

**Soundness:** 2
**Presentation:** 3
**Contribution:** 2
**Rating:** 3
**Confidence:** 4

**Summary:**

This paper proposes a method to distinguish between machine-generated and human-authored texts based on "entailment progressions", which is the textual entailment relation sequential changes between sentences within a text. Apart from evaluating the approach on existing datasets, the authors also present a new dataset.

The paper itself is quite clear and easy to understand. However, the underlying assumptions are unclear to me. In order to prove the effectiveness of this approach, two parts need to be clarified: 1) human-written text and machine-generated text are inherently differently in terms of entailment progression; and 2) the model needs to accurately recognize the entailment progression. I would suggest the authors to provide either empirical evidence or theoretical justification for 1) and evaluation results of the model for 2).

**Strengths:**

* Tackle the important topic of differentiating human-written text and machine-generate text
* Propose an approach to differentiate them based on discourse/logic structure underlying the text

**Weaknesses:**

* Lack of enough research on the field of recognizing textual entailment and discourse analysis of the text: for example,
  * Hickl, A. 2008. Using discourse commitments to recognize textual entailment. In COLING.
  * Wang, R., & Sporleder, C. 2010. Constructing a Textual Semantic Relation Corpus Using a Discourse Treebank. In LREC.
* Overly simplified modeling of the discourse/logic structure of the text: As the authors also mentioned it as "a heuristic", discourse relation is much more than entailment/contraction/neutral
* Lack of baseline, ablation tests, error analysis in the experiments: I would suggest to add basic classification-based method with language model scores, word embeddings, or other textual features, without entailment progression information. Otherwise, it is difficult to judge the effectiveness of the approach.

**Questions:**

See above

---

### Meta-Review · Area_Chair_k8P4 · 2024-12-06

**Metareview:**

This paper presents an idea of "entailment progressions": entailment values can be computed treating each sentence as a hypothesis and the preceding sentence(s) as a premise. A metric is proposed to detect machine-generated text based on this principle. While the idea here is intriguing, the work is underdeveloped both theoretically (reviewer tGAd particularly points out lack of engagement with prior work in discourse to motivate why this might be a good idea) and experimentally.

**Additional Comments On Reviewer Discussion:**

There was no rebuttal.

---

### Decision · Program_Chairs · 2025-01-22

Reject